# Crowdsensing Influences and Error Sources in Urban Outdoor Wi-Fi Fingerprinting Positioning

**DOI:** 10.3390/s20020427

**Published:** 2020-01-12

**Authors:** Cristian-Liviu Leca, Ioan Nicolaescu, Petrica Ciotirnae

**Affiliations:** Military Technical Academy ‘Ferdinand I’, 050141 Bucharest, Romania; ioannic@mta.ro (I.N.); petrica.ciotirnae@mta.ro (P.C.)

**Keywords:** crowdsensing, databases, smartphones, urban positioning, Wi-Fi fingerprinting

## Abstract

Wi-Fi fingerprinting positioning systems have been deployed for a long time in location-based services for indoor environments. Combining mobile crowdsensing and Wi-Fi fingerprinting systems could reduce the high cost of collecting the necessary data, enabling the deployment of the resulting system for outdoor positioning in areas with dense Wi-Fi coverage. In this paper, we present the results attained in the design and evaluation of an urban fingerprinting positioning system based on crowdsensed Wi-Fi measurements. We first assess the quality of the collected measurements, highlighting the influence of received signal strength on data collection. We then evaluate the proposed system by comparing the influence of the crowdsensed fingerprints on the overall positioning accuracy for different scenarios. This evaluation helps gain valuable insight into the design and deployment of urban Wi-Fi positioning systems while also allowing the proposed system to match GPS-like accuracy in similar conditions.

## 1. Introduction

Mobile phone sensing has recently attracted a great deal of attention due to factors such as their ubiquitous presence in everyday life, the addition of sensors not specific to their original design (barometers, accelerometers, light sensors, etc.), allowing them to be used intuitively for mobile sensing. Additionally, their permanent connection to the Internet using mobile data or a Wi-Fi connection has allowed the use of mobile phone sensors in areas such as transportation, tourism, logistics, air quality monitoring, and social networking [1,2]. Mobile phone sensing can have its powers augmented by engaging a large number of participants (a *crowd*) in contributing for a small cost or toward a common cause (e.g., to monitor road traffic and obtain real-time navigation) by the process of mobile crowdsensing.

We consider that the development of the mobile crowdsensing paradigm is essential to the development of functional Wi-Fi fingerprinting positioning systems for urban and outdoor areas. Previous research on fingerprinting positioning is mainly aimed at designing techniques and systems for indoor areas [3,4].

While the development of indoor location-based services and positioning systems have been motivated by complex, non-line-of-sight (NLOS) environments and the general unavailability of Global Positioning System (GPS) localization, urban crowdsensed fingerprinting positioning will compete in offering accurate positioning with the Global Navigation Satellite System (GNSS) for mobile devices equipped with both types of sensors and will prove to be a problem-solver for energy- and size-constrained IoT devices requiring GPS-like accurate positioning.

Reference [5], cited on the official *gps.gov* website, states that GPS has an average positioning accuracy of 5 m in rural areas and approximately 16.8 m in urban environments. This sets a clear accuracy goal for Wi-Fi positioning systems designed for urban environments, consisting of attaining similar or better accuracy with lower costs.

Early research on urban wireless positioning aimed at improving location-sensing accuracy by eliminating typical non-line-of-sight errors while not acknowledging the benefits of applying crowdsensing in enabling a Wi-Fi urban positioning system [6,7,8].

Other technologies that are readily available in smartphone devices used for crowdsensing, such as Bluetooth [9], cellular [10], accelerometers [11], magnetic sensors [12], or pedometers [13], can also be integrated in fingerprinting positioning systems as has been previously shown.

In this paper, we apply mobile crowdsensing to prove the feasibility of designing a Wi-Fi fingerprinting positioning system for urban areas. The envisioned system benefits from the ubiquitous presence of Wi-Fi access points (APs) in urban areas, as proven in our previous research [14,15,16]. The Wi-Fi AP received signal strength (RSS) measurements are made with the help of the 802.11n network interface of the mobile phones used for crowdsensing. The measurements used for positioning are coupled with the geographical position where they were captured, thus constructing the fingerprint database. The position of the measurements is known using the GNSS receivers of the smartphones used for crowdsensing.

We propose the usage of the crowdsensing technique as opposed to previous research using automated or grid-based fingerprint collection for indoor positioning systems. Finally, the envisioned positioning system will require continuous feedback from users that choose to contribute to the crowdsensing effort. This will enable:Continuous growth and resolution of the coverage area for the positioning systems;Coverage of areas restricted from public access (e.g., private homes);Constant feedback on the APs situation, allowing for the removal of inactive, mobile, or moved APs;A high density of fingerprints in areas most frequented by users, resulting in a higher accuracy of the positioning in areas most likely to be required.

Regarding the aspect of using different smartphones, our research was not aimed at mitigating the effects of distinct hardware and software, as this would prove to be an impossible task for a crowdsensing system. This is caused by the numerous smartphone devices available on the market that would ultimately be able to contribute to the proposed system.

The main challenge specific to the proposed solution lies in proving that fingerprinting positioning can achieve GPS-like accuracy in similar conditions specific to the urban environment. To address this we thoroughly evaluated the accuracy of the fingerprinting positioning system using different sub-datasets extracted from our collected data. For the position estimation, we apply the methods proposed in the papers published by Lohan et al. and Mendoza et al. [17,18].

The remainder of the article is organized as follows: Section 2 covers related work on similar positioning systems and highlights our previous work on the subject. Section 3 introduces the data collection method and the motivation behind choosing a specific test area for our research and discusses the implications of the crowdsensing technique on fingerprint database generation. In Section 4 we present the results of our analysis, highlighting the characteristics of the positioning algorithms used and the influence of the crowdsensed measurements. Section 5 concludes our work and highlights the main contributions of the paper.

## 2. Related Work

Outdoor positioning is usually based on multilateration or triangulation techniques [19] which work better in line-of-sight propagation conditions. These types of methods are not suited for urban environments affected by fading and NLOS propagation [15]. The approach presented in [20] aims at estimating the AP position with the help of information contained in the AP’s name. Indoor positioning systems [21] have also tackled the NLOS problem, one particular promising solution being the process of signal association with particular locations [22,23,24,25]. In this class of fingerprinting positioning methods, a position is solely characterized by its measured signal pattern, consisting of the consecutive collected Wi-Fi measurements. Thus, our proposed crowdsensed fingerprinting scheme is applied similarly without the need of knowing exact AP positions. Our method does not require distance or angle measurements, leading to its suitability for use in urban areas.

Wi-Fi fingerprinting has traditionally exploited Wi-Fi interfaces equipped on mobile devices, not limited to smartphones [26], as well as the ubiquitous presence of Wi-Fi APs [27]. Many of the early proposed systems [28,29] rely on an initial training phase required for constructing the fingerprint database to be used for the positioning phase. Outdoor positioning systems need significant time and effort for the training phase, which crowdsensing systems aim to limit. Crowdsensed systems are suited for outdoor environments where smartphones automatically collect RSS measurements coupled with GPS obtained locations for the training phase. As an alternative, crowdsourced systems have been proposed for indoor systems which require an active contribution from the user. In crowdsourced systems, the users are required to input their exact position in the positioning area using a dedicated app [30,31,32,33,34].

Designing a fingerprinting positioning system for urban environments requires knowledge of the presence and density of Wi-Fi APs in the system’s coverage area. Our previous work on fingerprinting positioning [15] proved the ubiquity of Wi-Fi APs in urban environments and offers insights regarding the data collection methodology, fingerprint database maintenance, and sources of error specific to the environment (e.g., mobile APs).

Achieving seamless urban positioning with no constraints on the indoor/outdoor environment conditions is a significant step towards implementing a global indoor positioning system, as proposed in [35].

As highlighted in previous research [17,18], Wi-Fi fingerprinting positioning systems encounter one major problem, consisting of challenges in implementing and reproducing results published in previous research papers on the subject. To address this issue, we formatted our data to match the proposed format for standard datasets in [17] and we used two of the positioning algorithms proposed in [18]—the weighted-centroid method and the log-Gaussian distance method.

## 3. Fingerprint Analysis and Test Data Description

The data collected for our research consisted of more than 1 million Wi-Fi measurements collected in Bucharest (Romania) and areas nearby the city by the help of eight different smartphones carried by different users for 6 months. The data collection process was a part of the everyday life of users and they were not required to scan in a specific way, space, or time. This means that the collected measurements were collected while stationary or when the users were moving from one place to another by any means available in the city.

Due to the crowdsensing method, the collected measurements are denser near the city center where the scanning occurred more often. This prompted us to select one of the most frequented areas, shown in Figure 1a during the crowdsensing step for the evaluation of the collected data on the accuracy of Wi-Fi fingerprinting positioning.

The chosen test area has a high density of Wi-Fi measurements and allows for long-term variations in Wi-Fi signal propagation. The area is characteristic to the urban environment of Romania, being placed between two high-traffic boulevards while the buildings in the area are tower apartment blocks. Locations of measurements in the test area are shown in Figure 1b.

The test area, with a surface of 1 km${}^{2}$, scanning resulted in 75,188 fingerprints of 10,072 distinct APs. The average signal level for all fingerprints was −83.85 dBm with a standard deviation of 5.46 dB.

The fingerprint coordinates were transformed from the decimal degree format to local coordinates. The decimal degree format is used to express latitude and longitude geographic coordinates as decimal fractions, while the local coordinates are referenced to the upper-left corner of the test area and are expressed in meters on the *x* and *y* axes.

The test area dataset and all subsequent sub-datasets were randomly split between training data (85% of the total) and test data (15% of the remaining fingerprints). The training fingerprints were used for estimating the position of the test fingerprints. The resulting estimated position was compared to the test fingerprint position resulting in a positioning error. Figure 2 is an example of training and test data positions with locally referenced coordinates expressed in meters.

Fingerprint collection by the crowdsensing technique has implications for the density and the received signal strength characteristics.

The density of fingerprints influences the accuracy of the crowdsensed positioning system and is closely linked to the number of people that have visited a certain area. Figure 3a displays the density of the gathered fingerprints using an interpolated graph overlaid on the test area map. Each colored dot signifies the presence of a number of APs in the area around it, the color of the dot ranging between blue (1 AP) and red (70 APs).

The average signal level (dBm) of the fingerprints in the test area is shown in Figure 3b. The signal map was generated using the interpolation technique with the average being calculated for a 10-m radius.

The placement of the AP, for example, the top of a building, and the crowdsensing technique influence the signal strength at ground level where the measurements were collected. APs placed on lower floors or at ground level have the chance to be scanned more often and be seen with a higher signal level, as shown in Table 1.

Figure 3c was obtained by plotting the density of APs scanned less than three times. By comparing Figure 3a,c we observe that APs scanned less than three times were observed at the margins of the test area where the overall density of measurements was lower. This leads to the following observations:

APs in densely scanned areas are themselves more likely to be scanned more often;The crowdsensing method offers consistent results for successive scans of the same area;APs’ radio visibility displays a character of stability for repeated scans.

Access points scanned less than three times can increase the positioning error significantly, as they are most likely placed at a great distance from the measurement points or are outside the test area.

## 4. Positioning Accuracy and Significant Error Sources Generated by Crowdsensing

The metric that used for the evaluation of the positioning accuracy of the crowdsensed data was the mean 2D error. The error was calculated as the Euclidean distance between the estimated position and the GPS-determined positions of the test measurements. The distances are expressed in meters using the local coordinates described previously.

As mentioned, the evaluation of the positioning accuracy in an urban test environment for the crowdsensed data was done using the weighted-centroid method [36,37] and the log-Gaussian distance method [18].

### 4.1. Weighted-Centroid Method

The weighted-centroid method was proposed for fingerprinting positioning in reference [37] as a simple and low-complexity approach. The estimated position is calculated as the weighted average of the AP positions heard in the unknown position.

Using the set of visible APs as H and the known AP positions as $c_{ap}\left(x_{ap},y_{ap}\right),\;ap=1,...,H$, the weighted-centroid position estimation is calculated as:(1)$${\hat{c}}_{MS,wc}=\frac{\sum_{ap\in H}w_{ap}c_{ap}}{\sum_{ap\in H}w_{ap}},$$
where $w_{ap}$ are weight functions.

Weighting is done according to the distance to the APs’ positions. Since the RSS can be used as a distance function to the APs, the weights can be replaced by the RSS, using the measurement vector $m_{n}={\left[m_{n,1},m_{n,2},...,m_{n,N_{ap}}\right]}^{T},\;n=1,...,N_{fp},$ where $N_{fp}$ is the number of measurements collected and $m_{(x,y),ap}$ is the RSS received from the AP ap at coordinates (x,y), resulting in the following RSS-based equation for the weighted-centroid method:(2)$${\hat{c}}_{MS,wc}=\frac{\sum_{ap\in H}m_{MS,ap}c_{ap}}{\sum_{ap\in H}m_{MS,ap}}.$$

Since our proposed technique is aimed at using crowdsensed measurements, the APs’ positions will not be known. Thus, the position of all measured APs was estimated as a first step using the weighted-centroid approach, similarly to the above steps.

### 4.2. Log-Gaussian Distance Method

The log-Gaussian distance method used in our paper was applied similarly as proposed in [38]. The position estimation was attained by the help of a weighted K nearest neighbor calculation on the most similar fingerprints stored in the database. The metric used to extract similar measurements to the one with the unknown position was based on the logarithmic Gaussian distance (LGD). The LGD between two measurement vectors p and r is:(3)$$LGD\left(p,r\right)=-\sum_{i}\log\max\left(G\left(p_{i},r_{i}\right),\epsilon \right),$$
where (G(p,r) is the Gaussian similarity between two values *p* and *r*, defined as: (4)$$G\left(p,r\right)=\left\{\begin{array}{cc} \frac{1}{\sqrt{2\pi \sigma^{2}}}\exp\left(-\frac{{\left(p-r\right)}^{2}}{2\sigma^{2}}\right), & \mathrm{if}\;p\neq 0\;and\;q\neq 0,\\ 0, & \mathrm{otherwise}. \end{array}\right.$$

### 4.3. Primary Results

The initial results for both algorithms when applied to the complete test area fingerprint database showed the following mean 2D error:30.017 m for the weighted centroid method;45.414 m for the log-Gaussian distance method.

These mean error values are significantly higher than those reported in previous studies where the test area had a smaller surface and was placed indoors, for example, approximately 10 m in [17] or approximately 6 m in [39]. When compared to previous results reported in papers evaluating positioning accuracy in urban environments, the attained results using crowdsensed data are similar, for example, reference [40] reports an average positioning accuracy between 23.5 and 36 m, while reference [41] reports errors between 35 and 120 m. When comparing indoor fingerprinting positioning research with outdoor fingerprinting positioning research results, we observe that the mean errors of outdoor systems are by an order of magnitude larger than those of indoor systems.

The mean error of outdoor fingerprinting positioning systems can be reduced using database processing, as proposed in [15,42,43], which publish results closing a 15-m average positioning error. The fingerprint database processing in the referenced papers involved eliminating the fingerprints of APs with an estimated coverage radius larger than 300 m and those of APs suspected to be mobile.

### 4.4. Characterization of the Positioning Algorithms

Each method has its characteristics which influence the resulting positioning error and also the way they benefit from the crowdsensed fingerprints. The implementation of the weighted-centroid method requires an initial step preceding the actual positioning that requires the estimation of the position of all the APs in the fingerprint database. Using crowdsensed data for AP position estimation can easily lead to errors in estimating the AP’s exact location. This is to be expected, as crowdsensed fingerprints are collected alongside roads or near buildings. The initial step mandates the study of the influence number of crowdsensed fingerprints for each AP.

The log-Gaussian method aims at finding the most similar fingerprints with the measurement vector used for positioning by the help of the log-Gaussian distance. The position estimation is then done using the *k*-nearest neighbor (*k*-NN) approach on the *k* most similar fingerprints. For the evaluation of the method, the number of neighbors was set to k=3. Varying the number of neighbors influences the results of the position estimation algorithm. For example, an increase in the value of *k* may lead to an estimated position further away from the test position due to the inclusion of less-similar fingerprints that were collected at a greater distance. This can be solved by applying a weight to the *k*-NN algorithm or by evaluating the influence of the crowdsensed data on the position estimation algorithm.

The particularities of each positioning method coupled with the less-known influence of the crowdsensed fingerprints prompted us to research the following aspects which influence the accuracy of crowdsensed fingerprinting positioning:The influence of mobile APs;The average number of fingerprints for each AP;The signal level of each AP measurement included in a fingerprint;The average signal level of the APs;The estimated coverage radius of the APs;The number of neighbors used for the *k*-NN estimation.

### 4.5. Influence of Mobile APs

Mobile APs can lead to significant positioning errors. For example, an AP installed in a public transport vehicle could lead to positioning estimation results placed in the part of the town where the AP was scanned. This prompted us to apply a method for removing the fingerprints of mobile APs, while also investigating the effects of their presence on the overall positioning performance.

The identification of mobile APs requires the analysis of the whole fingerprint database, not only of the test area. Mobile APs are classified using a method previously published by the authors in [15].

The method is based on the assumption that APs maximum coverage radius is rarely larger than 300 m. All measurement points for a distinct AP are selected and used to create a cluster. The minimum bounding box and the centroid point of the cluster are calculated. The maximum coverage of the AP is determined to be as the maximum distance between the centroid point and the minimum bounding box. Thus, APs with a range greater than 300 m are considered mobile.

By applying the described method we attained the following results:3506 APs representing 34.8% of the total were scanned once and could not be classified as fixed or mobile. Thus they were eliminated from the comparative dataset;395 APs representing 3.9% of the total were classified as mobile and were eliminated from the comparative dataset.

Table 2 displays the resulting average positioning error for both positioning methods when applied to the dataset with mobile APs removed. The weighted-centroid method displayed a minor increase in the average positioning error, caused by the elimination of APs seen only once while the log-Gaussian method benefited from the removal of the mobile APs.

### 4.6. Influence of the Average Number of Fingerprints Collected for Each AP

The average number of fingerprints per AP could influence both positioning methods when considering crowdsensing as the collection method. A low number of fingerprints can limit the information available in the AP position estimation step characteristic to the weighted-centroid method while a large number of fingerprints correlated to a large coverage area might reduce the unique character of each fingerprint, leading to an increase in the positioning error for both methods.

To study the previous hypothesis, the dataset was split into subsets using the average fingerprint number for each AP as the criteria as follows:Reduced fingerprint dataset containing APs averaging 1 to 5 fingerprints resulting in 13,647 fingerprints of 6798 APs;Average fingerprint dataset containing APs averaging 6 to 20 fingerprints resulting in 24,890 fingerprints of 2318 APs;Dense fingerprint dataset containing APs averaging 21 to 60 fingerprints resulting in 26,957 fingerprints of 828 APs;Very dense fingerprint dataset containing APs averaging more than 61 fingerprints resulting in 10,234 fingerprints of 128 APs.

The resulting mean positioning error for each sub-dataset is shown in Table 3.

The weighted centroid displayed increased precision for the reduced fingerprint dataset. This is caused by a higher positioning accuracy in scenarios where a reduced number of fingerprints were collected for the same APs in close proximity to each other. This is the case for APs that are visible in a narrow area at street level due to their placement in a distant position. For the datasets with a higher fingerprint-to-AP ratio, the precision was lower, mainly due to a decrease in the AP position estimation caused by the higher number and wider geographical distribution of the APs’ fingerprints.

The log-Gaussian method had a worse result for the reduced and very-dense datasets and significantly better accuracy for the intermediate values. A low number of fingerprints for APs means that less data is available for the statistic characterization of the data, while a large number of fingerprints often leads to the phenomena of geographic diffusion of precision where similar signal levels are found in fingerprints placed far away from each other.

Very dense fingerprint-to-AP ratios led to a decrease in positioning accuracy, leading to the conclusion that collection and storage of large volumes of fingerprints is not always beneficial to the fingerprint positioning system.

### 4.7. Influence of Fingerprint Signal Level

The signal level’s influence on positioning accuracy was studied by splitting the dataset into subsets using the signal level value expressed using dBm units as a threshold. The subset split aimed at keeping the number of fingerprints in each subset similar. The results of the analysis are displayed in Table 4.

As shown in Table 4, the log-Gaussian method’s accuracy was significantly lower than that of the full dataset. This was due to the usage of the log-Gaussian distance for determining the similarity between fingerprints and positioning measurements. When splitting the measurements using 5 dBm intervals it becomes more difficult for the method to properly identify similarities and differences between the fingerprints and unknown position measurements, leading to an increase in positioning error. The weighted-centroid method displayed increasing accuracy for signal levels above −80 dBm, as is to be expected for distance-based positioning methods.

### 4.8. Influence of Average AP Signal Level

The average AP signal level was calculated as the average level of all collected fingerprints for each distinct AP. Again, the dataset was split into subsets using the average signal level as the criteria, aiming at keeping the number of fingerprints in each set similar. The results of the analysis are displayed in Table 5.

As fewer APs were included in each subset due to the filtering, the weighted-centroid method performed worse in each scenario. The log-Gaussian method had a minor accuracy improvement for APs with an average signal level close to the average signal level of all collected fingerprints.

### 4.9. Influence of Estimated AP Coverage Radius

The AP coverage radius was estimated by the help of the crowdsensing process. The scanning process continued for five months inside the test area, allowing us to assume that street-level coverage was captured with precision. The average coverage radius of APs in the test area was approximately 47 m. The coverage area was determined by the help of the convex hull operation. The convex hull or convex envelope of a set of points *X* in a Euclidean plane or space is the smallest convex set that contains *x*. When applying the convex hull operation to the set of points consisting of the positions of all fingerprints collected for an AP, the result will be an estimation of the area covered by the AP.

The dataset was split into subsets using the estimated coverage radius as the criteria, and the results of the accuracy analysis are displayed in Table 6. The subset split aimed at keeping a similar number of fingerprints for each subset. This was possible for APs with a coverage radius under 80 m. APs with estimated coverage radius larger than 80 m were less numerous in the test area.

The resulting subsets based on the coverage radius had the following dimensions:Null radius—6506 fingerprints of 4988 APs;Radius between 0 and 30 m—17,618 fingerprints of 2634 APs;Radius between 30 and 50 m—19,603 fingerprints of 1328 APs;Radius between 50 and 80 m—18,659 fingerprints of 774 APs;Radius between 80 and 110 m—8074 fingerprints of 250 APs;Radius between 110 and 170 m—3665 fingerprints of 81 APs;Radius larger than 110 m—4728 fingerprints of 98 APs;

The positioning performance of the weighted-centroid method benefited from the usage of APs with an estimated coverage radius under 30 m, due to its initial step which requires the estimation of the AP position. A lower-radius AP is less likely to be positioned erroneously.

The log-Gaussian method had significantly greater accuracy for APs with an estimated radius under 50 m. For both methods, an AP coverage radius larger than 80 m resulted in larger positioning errors related to the geographic dilution of precision phenomena.

This analysis allows us to gain insight that can help set an upper limit on the estimated AP coverage area which in turn can limit the volume and effort related to the crowdsensing process.

### 4.10. Fingerprinting Filtering Summary

As previous filtering shows mixed results for the compared positioning methods, we highlight the results of the weighted-centroid and log-Gaussian methods in Table 7 and Table 8 by sorting them according to the 2D positioning error.

Table 7 shows that the weighted-centroid method benefited from using fingerprints collected from APs with a radius under 30 m and with signal levels between −71 and −80 dBm, meaning they were collected in the proximity of the AP.

Similarly to the weighted-centroid method, as Table 8 shows, the log-Gaussian method also benefited from APs with a low coverage radius. Aside from that, the log-Gaussian was characterized by increased accuracy for denser fingerprint subsets than the weighted-centroid method.

As can be seen in both tables, the methods had opposite performances for the same fingerprint subsets, except for the one with APs having a radius between 0 and 30 m. This results in the opportunity of using multiple estimation methods for the envisioned urban fingerprinting positioning systems that can be applied to different areas to be covered by the system. For example, if computed AP statistics show that an area is scanned more densely during the crowdsensing process, the log-Gaussian method should be used, as it outperformed the weighted-centroid method (38.080 m vs. 47.490 m average 2D error). Otherwise, if an area is characterized by a low average number of fingerprints for each AP the weighted-centroid method will outperform the log-Gaussian method (26.916 m vs. 91.198 m average 2D error).

### 4.11. Influence of the Number of Neighbors Specific to the *k*-NN Method

The *k*-NN method is often used for position estimation by fingerprinting systems. The *k*-NN method was combined with the log-Gaussian distance to select the most similar neighbors in the fingerprint space. Previous papers have proposed different values for *k*, according to the environment that the fingerprinting positioning system was to cover. References [28,44] propose that *k* = 3 or 4 for indoor fingerprinting systems.

The analysis of the influence of the number of neighbors on the accuracy of the positioning was done by varying the number between 1 and 10 and by computing the average 2D positioning error in each scenario. The dataset used contains 15,002 fingerprints of 1929 APs in the test area with an average estimated radius between 0 and 30 m. The results are displayed in Figure 4.

According to Figure 4, the average positioning error was smallest for k=2. This result is explained mainly by the data collection method. Collecting Wi-Fi measurements using crowdsensing means that most fingerprints will be captured alongside public roads or in the proximity of buildings, while most APs will be positioned indoors. Using three or more neighbors in the fingerprint space will likely lead to an estimation that is biased towards the two closest fingerprints, and as Figure 4 shows, farther than the real position of the user. When k=2 the positioning error was the smallest due to the probability that the real position was alongside the road on which the two fingerprints were collected.

## 5. Conclusions

Our analysis shows that the positioning accuracy of outdoor fingerprinting systems can be significantly improved by filtering and optimizing the contents of the fingerprint database. The best results came close to an average 2D positioning error of 21 m for the log-Gaussian distance method. These values do not come close to that achieved by previously published research on indoor systems. Nevertheless, our results show that urban fingerprinting positioning can reach GPS-like accuracy in similar use-cases as reported by reference [5].

The accuracy of crowdsensed fingerprinting positioning systems is shown to be dependent on the density of fingerprints, quantified as the average number of fingerprints collected for each AP. As the average positioning error was computed for the whole test area, including areas that were scanned less often and dense, it is safe to assume that for some of the most-scanned roads inside the test area the precision of the fingerprinting system would outperform that of GPS.

Wi-Fi fingerprint positioning systems for urban environments have become an intense topic of research in the context of Internet-of-Things technology. These positioning systems have the ability to offer GPS-like positioning accuracy for energy-constrained devices equipped with a Wi-Fi or 5G radio network interface that is used both for communications and positioning.

To evaluate the performance of urban fingerprint positioning systems, we built our fingerprint database using the crowdsensing technique for data gathering. Since the crowdsensing approach makes it difficult to cover a whole city with a small number of contributors, we chose to conduct our analysis on a dedicated test area specific to the Bucharest urban environment.

We analyzed the crowdsensed fingerprint and highlighted important aspects related to the research problem, including the presence of mobile APs, the influence of the average number of fingerprints collected per AP, the fingerprint signal level, the average street-level AP signal level, the influence of the estimated AP coverage radius, and the performance of the *k*-NN method.

We showed that the chosen positioning methods mostly performed in different ways when particular filters were applied. For example, the weighted-centroid method had a 17.922 m average 2D error for the subset with fingerprints had signal levels between −71 and −75 dBm, while the log-Gaussian method offered poor results at an average 2D error of 122.574 m. The main differences in the methods’ performances can be assigned to the fact that the weighted-centroid method requires an initial estimation of all of the APs’ positions, while the log-Gaussian method uses a similarity function that can perform poorly in particular situations, as proven by the fingerprint filtering.

We conclude by proving that fingerprint positioning systems can achieve or even out-perform satellite positioning systems in an urban environment, mostly due to the density of Wi-Fi APs and the difficulties that other positioning systems have in these conditions.

## Figures and Tables

**Figure 1 sensors-20-00427-f001:**
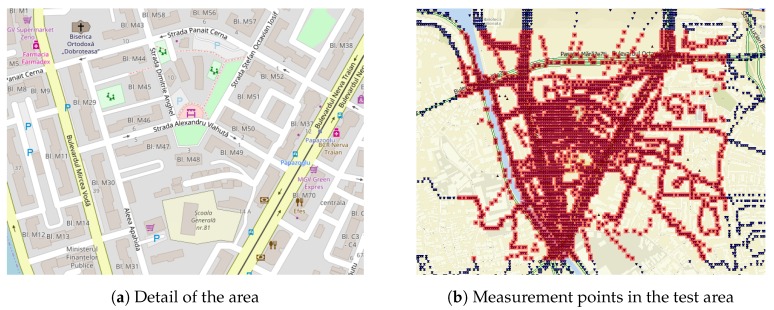
Test area overview.

**Figure 2 sensors-20-00427-f002:**
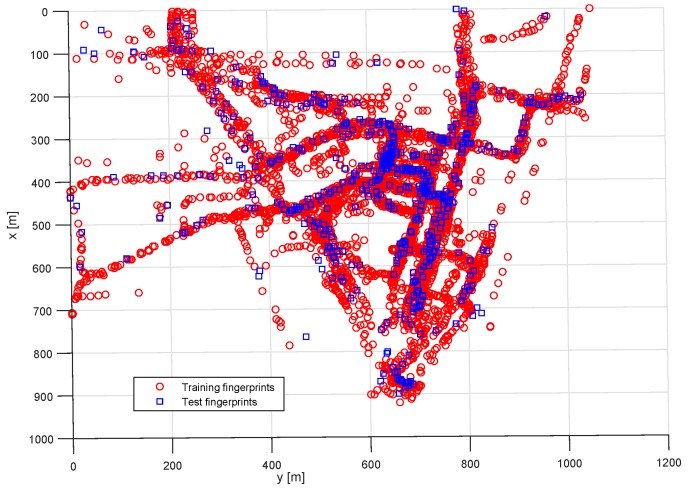
Training and test fingerprint positions.

**Figure 3 sensors-20-00427-f003:**
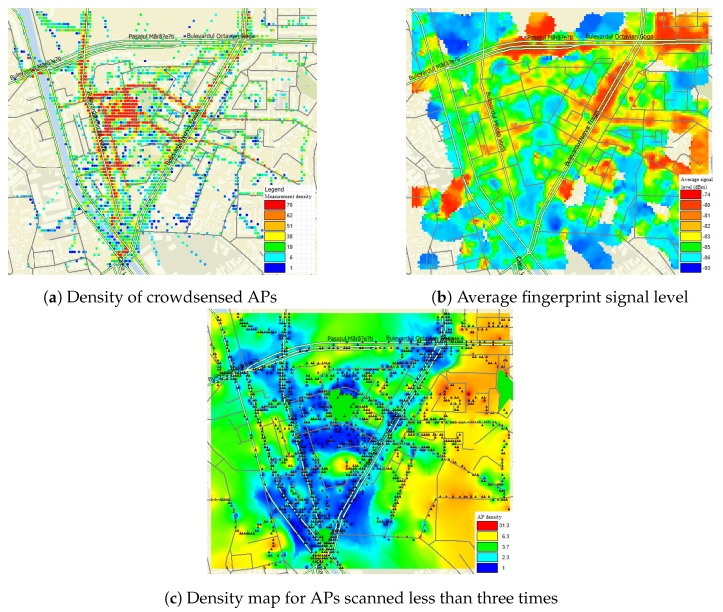
Test area analysis.

**Figure 4 sensors-20-00427-f004:**
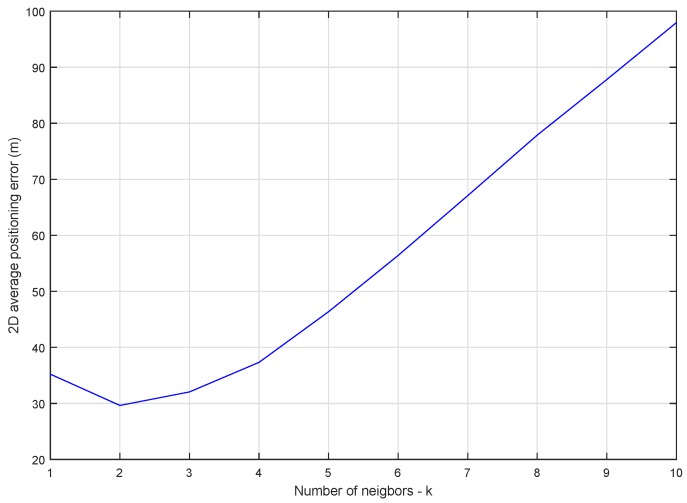
*k*-Nearest neighbor (*k*-NN) average 2D positioning error influenced by the number of neighbors.

**Table 1 sensors-20-00427-t001:** Frequency of measurements for the APs in the test area.

Frequency	APs Measured
1	3507
2	1446
3	791
4	575
5	479
6	384
7	297
8	229
9	202
10–19	1137
20–29	492
30–39	232
40–49	108
50–59	60
60–69	50
70–79	34
80–89	21
90–99	10
100–109	11
>110	7

**Table 2 sensors-20-00427-t002:** 2D positioning error for the dataset without mobile APs.

Dataset	Weighted-Centroid Mean Error (m)	log-Gaussian Mean Error (m)
Full dataset	30.017	45.414
Mobile AP removed set dataset	34.245 (−14%)	28.180 (+37%)

**Table 3 sensors-20-00427-t003:** 2D positioning error influenced by the average fingerprint-to-AP ratio.

Dataset	Weighted-Centroid Mean Error (m)	Log-Gaussian Mean Error (m)
Full dataset	30.017	45.414
Reduced fingerprint dataset	26.916 (+13%)	92.198 (−103%)
Average fingerprint dataset	37.896 (−20%)	31.147 (+30%)
Dense fingerprint dataset	49.439 (−64%)	38.274 (+15%)
Very dense fingerprint dataset	76.055 (−153%)	65.929 (−45%)

**Table 4 sensors-20-00427-t004:** 2D positioning error influenced by fingerprint signal level.

Dataset	Weighted-Centroid Mean Error (m)	Log-Gaussian Mean Error (m)
Full dataset	30.017	45.414
Subset under −90 dBm	32.394 (−8%)	118.566 (−161%)
Subset between −86 and −90 dBm	37.574 (−25%)	78.873 (−74%)
Subset between −81 and −85 dBm	31.318 (−4%)	71.872 (−58%)
Subset between −76 and −80 dBm	24.784 (+17%)	82.650 (−81%)
Subset between −71 and −75 dBm	17.922 (+40%)	122.574 (−169%)
Subset over −70 dBm	15.489 (+48%)	136.788 (−201%)

**Table 5 sensors-20-00427-t005:** 2D positioning error influenced by average AP signal level.

Dataset	Weighted-Centroid Mean Error (m)	Log-Gaussian Mean Error (m)
Full dataset	30.017	45.414
Subset under −90 dBm	32.394 (−8%)	118.566 (−161%)
Subset between −86 and −90 dBm	36.922 (−23%)	46.869 (−3%)
Subset between −81 and −85 dBm	42.766 (−42%)	40.885 (+10%)
Subset between −76 and −80 dBm	46.991 (−56%)	68.903 (−51%)
Subset between −71 and −75 dBm	32.799 (−9%)	166.648 (−266%)

**Table 6 sensors-20-00427-t006:** 2D positioning error influenced by average AP radius.

Dataset	Weighted-Centroid Mean Error (m)	Log-Gaussian Mean Error (m)
Full dataset	30.017	45.414
Null radius	11.033 (+63%)	278.320 (−512%)
Radius between 0 and 30 m	24.385 (+18%)	21.428 (+47%)
Radius between 30 and 50 m	33.744 (−12%)	28.127 (+38%)
Radius between 50 and 80 m	47.480 (−58%)	38.080 (+16%)
Radius between 80 and 110 m	69.585 (−131%)	57.914 (−27%)
Radius between 110 and 170 m	94.099 (−213%)	75.373 (−65%)
Radius larger than 110 m	111.839 (−272%)	97.662 (−115%)

**Table 7 sensors-20-00427-t007:** Fingerprinting filtering summary of the 2D positioning error for the weighted-centroid method.

Dataset	Weighted-Centroid Mean Error (m)	Log-Gaussian Mean Error (m)
Null radius	11.033 (+63%)	278.320 (−512%)
Fingerprint level between −71 and −75 dBm	17.922 (+40%)	122.574 (−169%)
Radius between 0 and 30 m	24.385 (+18%)	21.428 (+47%)
Fingerprint level between −76 and −80 dBm	24.784 (+17%)	82.650 (−81%)
Reduced fingerprint dataset	26.916 (+13%)	92.198 (−103%)
Full dataset	**30.017**	**45.414**
Fingerprint level between −81 and −85 dBm	31.318 (−4%)	71.872 (−58%)
Radius between 30 and 50 m	33.744 (−12%)	28.127 (+38%)

**Table 8 sensors-20-00427-t008:** Fingerprinting filtering summary of the 2D positioning error for the log-Gaussian method.

Dataset	Weighted-Centroid Mean Error (m)	Log-Gaussian Mean Error (m)
Radius between 0 and 30 m	24.385 (+18%)	21.428 (+47%)
Radius between 30 and 50 m	33.744 (−12%)	28.127 (+38%)
Average fingerprint dataset	37.896 (−20%)	31.147 (+30%)
Radius between 50 and 80 m	47.480 (−58%)	38.080 (+16%)
Dense fingerprint dataset	49.439 (−64%)	38.274 (+15%)
Average AP signal level between −81 and −85 dBm	42.766 (−42%)	40.885 (+10%)
Full dataset	**30.017**	**45.414**
Average AP signal level between −86 and −90 dBm	36.922 (−23%)	46.869 (−3%)

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
