# Peer review of "Crowdsensing Influences and Error Sources in Urban Outdoor Wi-Fi Fingerprinting Positioning"

_sensors, 2020, doi:10.3390/s20020427_

Round 1

Reviewer 1 Report

This is an empirical work based on a substantial  data set. While the paper does not introduce significant novel techniques, the work presented is quite thorough and informative to researchers in the area.

Author Response

Response to Reviewer 1 Comments

On behalf of all authors, I state that the work is primarily aimed at proving the feasibility of applying known methods for fingerprinting positioning in the urban environment.

Thank you for your review of our paper.

Reviewer 2 Report

The authors propose methods to improve the quality of the crowd-sensed Wifi fingerprint collected outdoors in an urban environment.

I am not sure why the authors are attempting to use WiFi fingerprinting outdoors, where GPS is available for relatively high-accuracy positioning. As the authors point out, crowdsensing requires labeling of positions using GPS with measured fingerprints, which beats the purpose of using WiFi fingerprints. If the authors have a motivation for the need for outdoor positioning without GPS, then they should clearly describe this. 

The authors also analyze several criteria for selecting measurements from the full set to improve the quality of the fingerprints. I am assuming that these will be ultimately used in combination to improve the overall estimation accuracy. However, the authors have not done any analyses with regard to combining the proposed selection criteria. 

Some specific comments follow:

p. 1, line 27

"urban crowdsensed fingerprinting positioning will compete with
28 GPS positioning for mobile devices" 

-> Again, what is your motivation/justification for this comment?

p. 6, line 137

"The error is calculated as the difference between the estimated position and the 138 known positions of the test data sets."

-> It should be clearly stated that the "known positions" are detected using GPS.

p. 6, line 140

The weighted-centroid method and the log-Gaussian distance method should be described here.

p. 6, line 142

Define "mean 2D error". Is this the Euclidean distance between the estimated and the know positions?

p. 6, line 157

Describe how the "coverage radius" is estimated, how APs are suspected to be mobile (based on what criteria), and how GPS positionings are determined to be inaccurate.

p. 7, line 182

Define "the signal level of fingerprints." Is this the average RSSI of all detected APs in a fingerprint?

p. 7, line 192

"Mobile APs are classified using a method previously published by the authors" -> Briefly describe this method here. 

p. 7, line 199

"The weighted-centroid method displays a minor increase in the average positioning error, caused by the elimination of APs seen only once" -> Why do you think so? 

p. 7, Section 5.2

This section gives mixed results according to the estimation methods. Should mobile APs be deleted, or retained?

p. 8, Table 2

"Comparative data-set" -> "Moble AP removed"

p. 8, line 220

"This is caused by a higher positioning accuracy in scenarios where the estimated position of the APs in the positioning measurement are placed closely." -> Elaborate. 

p. 8, line 222

"decrease in the AP position estimation" -> decrease in the estimation error?

"decrease in the AP position estimation caused by the higher number of fingerprints." -> Why did you come to this conclusion?

p. 8, line 233

"value of the signal level in dBm" -> Elaborate. Do you mean to select only the RSSI measurements that are above a defined threshold?

p. 8, line 235

"the log-Gaussian method accuracy is significantly lower than that of the full data-set. This is due to the usage of the log-Gaussian distance for determining the similarity between fingerprints and positioning measurements." Why? How did you come to this conclusion?

p. 9, line 239

"insights" ->What specific "insights"?

p. 9, line 242

"average level of all collected fingerprints of each AP"

Elaborate. 

p. 9, line 250

"The AP coverage radius is estimated by the help of the crowdsensing process." ->How? How are the AP positions estimated?

Author Response

Response to Reviewer 2 Comments

The authors propose methods to improve the quality of the crowd-sensed Wifi fingerprint collected outdoors in an urban environment.

I am not sure why the authors are attempting to use WiFi fingerprinting outdoors, where GPS is available for relatively high-accuracy positioning. As the authors point out, crowdsensing requires labeling of positions using GPS with measured fingerprints, which beats the purpose of using WiFi fingerprints. If the authors have a motivation for the need for outdoor positioning without GPS, then they should clearly describe this. 

Response:

We consider that Wi-Fi positioning systems have the potential to surpass GNSS positioning accuracy in dense urban environments enabling location based services for energy constrained devices which would not need be equipped with both Wi-Fi and GNSS sensors. We also consider for future research expanding our positioning system to 5G NR measurements.

Text edit: Reference \cite{van2015worlds}, cited on the official \textit{gps.gov} website states that GPS has an average positioning accuracy of 5 meters in rural areas and approximately 16.8 meters in urban environments. This sets a clear accuracy goal for Wi-Fi positioning systems designed for urban environments consisting of attaining similar or better accuracy with lower costs.

The authors also analyze several criteria for selecting measurements from the full set to improve the quality of the fingerprints. I am assuming that these will be ultimately used in combination to improve the overall estimation accuracy. However, the authors have not done any analyses with regard to combining the proposed selection criteria. 

Response:

The combination of the criteria and the trimming of the fingerprint database is a current topic of research for the authors as we acknowledge the potential benefits on the overall accuracy and precision of the positioning system.

Some specific comments follow:

Point 1 : p. 1, line 27

"urban crowdsensed fingerprinting positioning will compete with
28 GPS positioning for mobile devices" 

-> Again, what is your motivation/justification for this comment?

Response 1:  The “competition” is that of offering the best positioning accuracy.

Text edit: While the development of indoor Location-based-services and positioning systems have been motivated by complex, non-line-of-sight(NLOS) environments and a general unavailability of GPS positioning, urban crowdsensed fingerprinting positioning will compete in offering accurate positioning with Global Navigation Satellite System (GNSS) for mobile devices equipped with both types of sensors and will really prove to be a problem-solver for energy and size constrained IoT Devices requiring GPS-like accurate positioning.

Point 2 : p. 6, line 137

"The error is calculated as the difference between the estimated position and the 138 known positions of the test data sets."

-> It should be clearly stated that the "known positions" are detected using GPS.

 Response 2:

Text edit: The error is calculated as the difference between the estimated position and the GPS-determined positions of the test measurements. 

6, line 140

The weighted-centroid method and the log-Gaussian distance method should be described here.

 Response 3:

The methods description was included in the paper.

6, line 142

Define "mean 2D error". Is this the Euclidean distance between the estimated and the know positions?

Response 4: 

Text edit: The metric that is used for the evaluation of the positioning accuracy of the crowdsensed data is the mean 2D error. The error is calculated as the Euclidean distance between the estimated position and the GPS-determined positions of the test measurements. The distances are expressed  in meters using the local coordinates described before.

6, line 157

Describe how the "coverage radius" is estimated, how APs are suspected to be mobile (based on what criteria), and how GPS positionings are determined to be inaccurate.

Response 5:

Text edit: The coverage area was determined by the help of the Convex Hull operation. The convex hull or convex envelope of a set of points $X$ in a Euclidean plane or space is the smallest convex set that contains $x$. When applying the convex hull operation to the set of points consisting of the positions of all fingerprints collected for an AP the result will be an estimation of the area covered by the AP.

Mobile APs are detected using a method previously published by the authors and referenced in the paper:

L. Leca, "Ubiquity of Wi-Fi: Crowdsensing Properties for Urban Fingerprint Positioning," Advances in Electrical and Computer Engineering, vol.17, no.4, pp.131-136, 2017, doi:10.4316/AECE.2017.04016

The methods description is included in the text:

The method is based on the assumption that APs maximum coverage radius is rarely larger than 300 meters. All measurement points for a distinct AP are selected and used to create a cluster. The minimum bounding box and the centroid point of the cluster are calculated. The maximum coverage of the AP is determined to be as the maximum distance between the centroid point and the minimum bounding box. Thus APs with a range greater than 300 meters are considered mobile.

The claim of eliminating inaccurate GPS positioning is mentioned as a reference to published research and was removed for clarity.

7, line 182

Define "the signal level of fingerprints." Is this the average RSSI of all detected APs in a fingerprint?

Response 6: It is the received signal strength indicator for each AP visible in the measurement point.

Text edit: the signal level of each AP measurement included in a fingerprint

7, line 192

"Mobile APs are classified using a method previously published by the authors" -> Briefly describe this method here. 

Response 7:

Text edit as also suggested in response 5

7, line 199

"The weighted-centroid method displays a minor increase in the average positioning error, caused by the elimination of APs seen only once" -> Why do you think so? 

 Response 8: The weighted-centroid method displays a minor increase in the average positioning error, caused by the elimination of APs seen only once due to the split of the database into training and test sub-sets. The split has the potential to increase the positioning error due to the fact that the weighted-centroid method implies an initial step involving AP position estimation. If the AP position estimation is done with less data this leads to a chance of larger errors. This issue is specific to the weighted-centroid method and could be addressed more specifically in a more detailed paper on the subject.

7, Section 5.2

This section gives mixed results according to the estimation methods. Should mobile APs be deleted, or retained?

 Response 9: As the results are mixed this suggests that the decision should be taken during the steps of choosing the final implementation of the proposed positioning system.

8, Table 2

"Comparative data-set" -> "Moble AP removed"

 Response 10. Text was edited as suggested.

8, line 220

"This is caused by a higher positioning accuracy in scenarios where the estimated position of the APs in the positioning measurement are placed closely." -> Elaborate. 

Response 11. Text rephrased:

The weighted centroid displays increased precision for the reduced fingerprint data-set. This is caused by a higher positioning accuracy in scenarios were a reduced number of fingerprints were collected for the same APs in close proximity to each other. This is the case for APs that are visible in a narrow area at street level due to their placement in a distant position. For the data-sets with a higher fingerprint to AP ratio, the precision is lower, mainly due to a decrease in the AP position estimation caused by the higher number and wider geographical distribution of the AP's fingerprints.

The described situation is caused by the crowdsensing at street level method for data collection

8, line 222

"decrease in the AP position estimation" -> decrease in the estimation error?

"decrease in the AP position estimation caused by the higher number of fingerprints." -> Why did you come to this conclusion?

 Response 12. We failed to include the full description, that is rephrased as: decrease in the AP position estimation caused by the higher number and wider geographical distribution of the AP's fingerprints.

8, line 233

"value of the signal level in dBm" -> Elaborate. Do you mean to select only the RSSI measurements that are above a defined threshold?

Response 13: The sub-sets contain measurements with values above and under the specified thresholds.

Text rephrase: The signal level influence on positioning accuracy was studied by splitting the data-set into sub-sets using the signal level value expressed using dBm units as a threshold

8, line 235

: "the log-Gaussian method accuracy is significantly lower than that of the full data-set. This is due to the usage of the log-Gaussian distance for determining the similarity between fingerprints and positioning measurements." Why? How did you come to this conclusion?

Response 14: The log-Gaussian method is based on the log-Gaussian distance between two vectors as a distance metric. When splitting the measurements using 5 dBm intervals it becomes more difficult for the algorithm to properly identify similarities and differences between the fingerprints and unknown position measurements leading to an increase in positioning error.

 Text edit: This is due to the usage of the log-Gaussian distance for determining the similarity between fingerprints and positioning measurements. When splitting the measurements using 5 dBm intervals it becomes more difficult for the method to properly identify similarities and differences between the fingerprints and unknown position measurements leading to an increase in positioning error.

9, line 239

"insights" ->What specific "insights"?

 Response 15: Text rephrase : The weighted-centroid method displays increasing accuracy for signal levels above -80 dBm as is to be expected for distance based positioning methods.

9, line 242

"average level of all collected fingerprints of each AP"

Elaborate. 

 Response 16: Rephrase: The average AP signal level is calculated as the average level of all collected fingerprints for each distinct AP

9, line 250

"The AP coverage radius is estimated by the help of the crowdsensing process." ->How? How are the AP positions estimated?

Response 17:

Rephrase and description of the method for AP coverage area calculation:

The AP coverage radius is estimated by the help of the crowdsensing process. The scanning process continued for five months inside the test area allowing us to assume that street-level coverage was captured with precision. The average coverage radius of APs in the test area is approximately 47 meters.

The coverage area was determined by the help of the Convex Hull operation. The convex hull or convex envelope of a set of points $X$ in a Euclidean plane or space is the smallest convex set that contains $x$. When applying the convex hull operation to the set of points consisting of the positions of all fingerprints collected for an AP the result will be an estimation of the area covered by the AP.

Thank you for your review of our paper.

Reviewer 3 Report

This paper deals with the error sources of urban WiFi positioning based on crowdsensing and provide experiment results with large number of measurements.

The paper seems to be beneficial and provide helpful information to related researchers.

1. Related to crowdsensing, it seems that only recent research works are cited.
To claim authors' contributions, more survey and inclusion of earler works are recommended for more references.
Parts of recommended references are:
- H. K. Lee, J. Y. Shim, H. S. Kim, B. Li, and C. Rizos, "Feature Extraction and Spatial Interpolation for Improved Wireless Location Sensing", Sensors, Vol. 8, pp. 2865-2885, 2008

2. In the paper, it is not clear written whether and how the refence fingerprint coordinates were utilized. If so, it would be also helpful whether authors used GPS coordinates.

3. More detailed explanations on the terminology 'decimal degree global format' and 'local coordinates' are required.

4. The authors should clarify whether the same type or different types of smartphons were used to collect WiFi measurements for crowdsensing.

5. Page6, Line 143, 144:
Check 30,017 => 30.017 ?
Check 45,414 => 45.414 ?

6. For improved and self-contained readability, add brief equations describing the weighted-centroid and log-Gaussian methods.

7. Page 9, Line 253:
More detailed explanations regarding the following sentence should be added. It is difficult to understand.
"The coverage area was determined by the help of the Convex Hull operation, specific to
254 database management systems."

8. The 3rd line of Table 4, 5, 6:
Check 30,017 => 30.017 ?
Check 45,414 => 45.414 ?

9. It is also unclear how the AP coordinated are estimated and utilized. Add more explanations on this.

10. Page 10, Line 279:
Typo error : 3or4 => 3 or 4 ?

Author Response

Response to Reviewer 3 Comments

Point 1: Related to crowdsensing, it seems that only recent research works are cited.
To claim authors' contributions, more survey and inclusion of earler works are recommended for more references.
Parts of recommended references are:
- H. K. Lee, J. Y. Shim, H. S. Kim, B. Li, and C. Rizos, "Feature Extraction and Spatial Interpolation for Improved Wireless Location Sensing", Sensors, Vol. 8, pp. 2865-2885, 2008

Response 1:

Text edit: Early research on urban wireless positioning aimed at improving location-sensing accuracy by eliminating typical non-line-of-sight errors while not acknowledging the benefits of applying crowdsensing in enabling a Wi-Fi urban positioning system \cite{lee2008feature,lee2003framework,lee2005implementation}.

We included references to previous research on urban wireless positioning.

Point 2: In the paper, it is not clear written whether and how the refence fingerprint coordinates were utilized. If so, it would be also helpful whether authors used GPS coordinates.

Response 2:

Text edit: The measurements used for positioning are coupled with the geographical position where they were captured thus constructing the fingerprint database. The position of the measurements is known using the Global Navigation Satellite System (GNSS) receivers of the smartphones used for crowdsensing.

Point 3. More detailed explanations on the terminology 'decimal degree global format' and 'local coordinates' are required.

Response 3:

Text edit: The fingerprints coordinates were transformed from the decimal degree format to local coordinates. The decimal degree format is used to express latitude and longitude geographic coordinates as decimal fractions, while the local coordinates are referenced to the upper left corner of the test area and are expressed in meters on the x and y axis.

Point 4. The authors should clarify whether the same type or different types of smartphons were used to collect WiFi measurements for crowdsensing.

Response 4:

Text edit: The data collected for our research consisted of more than 1 million Wi-Fi measurements collected in Bucharest(Romania) and areas nearby the city by the help of 8 different smartphones carried by different users for 6 months.

Point 5. Page6, Line 143, 144:
Check 30,017 => 30.017 ?
Check 45,414 => 45.414 ?

Response 5:

The decimal separator was corrected.

Point 6. For improved and self-contained readability, add brief equations describing the weighted-centroid and log-Gaussian methods.

Response 6: The methods were described in the paper.

Page 9, Line 253:
More detailed explanations regarding the following sentence should be added. It is difficult to understand.
"The coverage area was determined by the help of the Convex Hull operation, specific to
254 database management systems."

Response 7:

Text edit:

The coverage area was determined by the help of the Convex Hull operation. The convex hull or convex envelope of a set of points $X$ in a Euclidean plane or space is the smallest convex set that contains $x$. When applying the convex hull operation to the set of points consisting of the positions of all fingerprints collected for an AP the result will be an estimation of the area covered by the AP.

The 3rd line of Table 4, 5, 6:
Check 30,017 => 30.017 ?
Check 45,414 => 45.414 ?

Response 8:

The decimal separator was corrected.

It is also unclear how the AP coordinated are estimated and utilized. Add more explanations on this.

Response 9. By including details on the estimation methods used in the paper as request in Point 6 we hope that it helps clarify how the APs coordinates are estimated and utilized

Page 10, Line 279:
Typo error : 3or4 => 3 or 4 ?

Response 10: The typo was corrected .

Thank you for your review of our paper.

Reviewer 4 Report

General remarks:

Chapters can and probably should be combined for more clarity, e.g. 3 & 4; 5 & 6 Minor formatting issues (e.g. couple of spaces are missing).

Disclaimer: I am very interested in this research field, but in no way an expert. Some of my suggestions might be obvious to colleagues/experts on the field, but could improve clarity for interested non-experts.

Concerning each chapter:

Chapter 1 and 2: the general concept of crowdsensing in the context of the proposed fingerprinting positioning could be further explained. E.g. is data akquisition/training only done using designated devices or also during regular operation What about other communication technologies? Cellular communication or Bluetooth? Is there work on combined performances of GNSS and Wi-Fi fingerprinting or is it even considerable? Line 88 and following: Considered positioning methods are mentioned, but not further explained in the paper. This should be added to help understanding possible positive and negative effects of different factors (as discussed in chapter 5). Chapter 3 and 4: This chapter should be extended. Since measuring RSS is highly dependant on various influences, it should be stated if 8 equal or different smartphones were used; if data akquisition was only performed during travelling or if longer duration of stays were included; have the phones been carried by pedestrians or vehicle drivers or all modes combined. How does the use of different phones with different communication modules/antennas influence RSS and therefore overall fingerprinting performance? Map figures (1, 2, 3, 4, 5 and 6) should be combined in subfigures, e.g. instead of 6 sole figure, have 3 subfigures without losing figure size. Table 1 should be reformatted. To me it was unclear how AP positions are estimated. A further explanation would be helpful. Same goes with applied positioning methods or mention relevant sources in more detail. Chapter 5: References to positioning accuracy from other works should be included in related work. Tables in this chapter could also be combined to give an overall overview. Lines 198-201: Observable effects are mentioned in the text, but require some sort discussion / reasoning. I am having trouble drawing conclusions from the proposed dataset filtering. Can you conclude any general validity from your observations? It seems like none of the data filtering approaches shows constant improvements for both positioning methods. This should be further discussed. Data sub-set selection could be explained in more detail. Could there be different aspects to consider rather than equal sub-set sizes? Also sub-set selection in lines 281-284. Chapter 6: This sounds more like a part of the conclusion. Either remove this chapter or have a deeper discussion on observable effects from chapter 5. Chapter 7: This should be connected closer to the actual paper, rather than just stating that crowdsensed Wi-Fi fingerprinting can outperform GNSS accuracies. Quantitative results should also be given in more detail.

Author Response

I attach the response to your suggestions. We thank you for the help provided in the review.

Best regards,

The authors

Round 2

Reviewer 4 Report

Thank you for your replies and excessive editing of the submission.